

# Metabolomics window into the role of acute kidney injury after coronary artery bypass grafting in diabetic nephropathy progression

Jiayi Wang[1], Wenzhe Yan[2], Xiang Zhou[1], Yu Liu[1], Chengyuan Tang[1], Youming Peng[1], Hong Liu[1], Lin Sun[1], Li Xiao[1] and Liyu He[1]

[1] Department of Nephrology, The Second Xiangya Hospital, Central South University, Key Lab of Kidney Disease and Blood Purification in Hunan, Changsha, China
[2] Department of Hematology, The Second Xiangya Hospital, Central South University, Changsha, China

Corresponding author
Liyu He, heliyu1124@csu.edu.cn

## ABSTRACT

**Introduction.** Metabolomics has emerged as a valuable tool to discover novel biomarkers and study the pathophysiology of diabetic nephropathy (DN). However, the effect of postoperative acute kidney injury (AKI) on diabetes mellitus (DM) to chronic DN progression has not been evaluated from the perspective of metabolomics.

**Methods.** A group of type 2 diabetes mellitus (T2DM) inpatients, who underwent off-pump coronary artery bypass grafting (CABG), were enrolled in our study. According to whether postoperative AKI occurred, patients were grouped in either the AKI group (AKI, $n = 44$) or the non-AKI group (NAKI, $n = 44$). Urine samples were collected from these patients before and 24 h after operation. Six patients from the AKI group and six patients from the NAKI group were chosen as the pilot cohort for untargeted metabolomics analysis, with the goal of identifying postoperative AKI-related metabolites. To understand the possible role of these metabolites in the chronic development of renal injury among T2DM patients, trans-4-hydroxy-L-proline and azelaic acid were quantified by targeted metabolomics analysis among 38 NAKI patients, 38 AKI patients, 46 early DN patients (DN-micro group), and 34 overt DN patients (DN-macro group).

**Results.** Untargeted metabolomics screened 61 statistically distinguishable metabolites in postoperative urine samples, compared with preoperative urine samples. Via Venn diagram analysis, nine of 61 were postoperative AKI-related metabolites, including trans-4-hydroxy-L-proline, uridine triphosphate, p-aminobenzoate, caffeic acid, adrenochrome, δ-valerolactam, L-norleucine, 5′-deoxy-5′-(methylthio) adenosine, and azelaic acid. By targeted metabolomics analysis, the level of trans-4-hydroxy-L-proline increased gradually from the NAKI group to the AKI, DN-micro, and DN-macro groups. For azelaic acid, the highest level was found in the NAKI and DN-micro groups, followed by the DN-macro group. The AKI group exhibited the lowest level of azelaic acid.

**Conclusions.** The detection of urinary trans-4-hydroxy-L-proline after AKI could be treated as an early warning of chronic DN progression and might be linked to renal fibrosis. Urinary azelaic acid can be used to monitor renal function noninvasively in DM and DN patients. Our results identified markers of AKI on DM and the chronic

progression of DN. In addition, the progression of DN was associated with AKI-like episodes occurring in DM.

## INTRODUCTION

Until 2017, about 451 million adults live with diabetes mellitus (DM) in the world. Moreover, 44.4% of the type 2 DM (T2DM) will progress to diabetic nephropathy (DN) in three decades. DN is now the most frequent cause of end stage renal disease (ESRD) (*Cho et al., 2018*; *Hallan et al., 2006*). Microalbuminuria is a useful diagnostic marker and major risk factor for DN progression; however, it has been challenged for its limited sensitivity and specificity (*Chen et al., 2017*; *Tabaei et al., 2001*). Multiple studies have explored better biomarkers for detecting early pathophysiologic changes of DN (*Colhoun & Marcovecchio, 2018*).

Metabolomics, the systems biology approach for the identification and quantification of global metabolites in biological samples (*Chen et al., 2018a*), has emerged as a valuable tool to discover novel biomarkers of DM and DN (*Arneth, Arneth & Shams, 2019*; *Darshi, Van Espen & Sharma, 2016*). Using metabolomics, significant changes in the serum level of leucine, dihydrosphingosine, and phytoshpingosine were noted in DM and DN patient groups, compared with the controls (*Zhang et al., 2009*). Metabolite profiling in participants from two large, longitudinal studies also identified five branched-chain and aromatic amino acids (isoleucine, leucine, valine, tyrosine, and phenylalanin) that are highly associated with future diabetes (*Wang et al., 2011*). Meanwhile, metabolomics helps to elucidate amino acid metabolism, phospholipid metabolism, and lipid metabolism, which may be the important biological mechanisms that promote DM and DN (*Rossi et al., 2018*; *Zhang et al., 2009*). Urine is a noninvasive sample source that contains all renal metabolic end products. Thus, urine metabolomics has been used for predicting the development of nephropathy in T2DM patients (*Chen et al., 2018a*), DN progression from micro- to macroalbuminuria (*Pena et al., 2014*), and ESRD risk of T2DM patients with microalbuminuria (*Tang et al., 2019*).

DN is rarely a static condition but rather one that evolves and changes over time during the lifespan of the individual. Acute kidney injury (AKI) could participate in and promote the dynamic progression of DN. It was reported that the yearly AKI incidence of T2DM was 198/100,000, versus 27/100,000 among nondiabetic individuals (*Girman et al., 2012*). In animal models, ischemic AKI was more severe in diabetic mice than in non-diabetic mice (*Gao et al., 2013*; *Peng et al., 2015*). These studies indicate that DM predisposes and sensitizes patients to AKI. However, the effect of AKI on DM to chronic DN progression has not been evaluated from the perspective of metabolomics.

AKI is one of the most important complications after coronary artery bypass grafting (CABG) procedure, especially for DM patients (*Bellomo et al., 2008*). Hence, we choose

T2DM patients undergoing elective off-pump CAGB as our main study objects to examine AKI-associated metabolites among DM patients. Untargeted metabolomics analysis was performed on the urine samples of these patients in the discovery stage. To further explore the role of identified postoperative AKI markers in the chronic development of renal injury among T2DM patients, the discovered metabolites were quantified by targeted metabolomics analysis among DM patients with or without AKI after CABG, DN patients with microalbuminuria, and DN patients with macroalbuminuria. The results suggest potential biomarkers for early detection and progression of DN, as well as associate chronic DN progression with AKI-like episodes. This helps to visualize the dynamic progress of DN via metabolomics.

## MATERIALS AND METHODS

### Definitions

T2DM was diagnosed via WHO criteria. Normoalbuminuria was defined as urinary albumin excretion rate (UAER) <30 mg/day; microalbuminuria was defined as UAER = 30–300 mg/day, and macroalbuminuria as UAER >300 mg/day. T2DM with microalbuminuria (DN-micro) was defined as early DN. T2DM with macroalbuminuria (DN-macro) was defined as overt DN. Postoperative AKI was defined as an absolute increase in serum creatinine by $\geq$0.3 mg/dL (26.5 µmol/L) from baseline within 48 h or $\geq$50% within 7 days after operation. The staging of AKI was evaluated according to the criteria proposed by Kidney disease: Improving global outcomes (KDIGO) classifications (*Kellum, Lameire & Group, 2013*). Anemia was defined as hemoglobin less than 110 g/L in females or 120 g/L in males. The definition of chronic kidney disease (CKD) followed the National Kidney Foundation-Kidney Disease Outcomes Quality Initiative (K/DOQI) guidelines, and the CKD stage was defined using patient baseline estimated glomerular filtration rate (eGFR) (*National Kidney Foundation, 2002*). eGFR was calculated using the Chronic Kidney Disease Epidemiology Collaboration equation (CKD-EPI).

### Patient recruitment and urine collection

T2DM patients who received the elective off-pump CABG procedure between Jan 2018 and Aug 2019 at the Second Xiangya Hospital of Central South University were screened for study participation. The inclusion criteria were as follows: (1) T2DM with normoalbuminuria; (2) Age ranged from 30 to 75 years; (3) eGFR >60 mL/min/1.73 m$^2$. Exclusion criteria were as follows: (1) Diagnosis with type 1 diabetes mellitus (WHO criteria); (2) Other etiology of CKD; (3) AKI prior to surgery; (4) Acute and chronic inflammation of the urinary system or other systems; (5) Patients with significant coexisting illnesses likely to affect survival or combined with other systemic diseases (including but not limited to malignancy, severe heart failure, blood system diseases, autoimmune diseases, cancer patients, and AIDS); (6) Patients with insufficient clinical data; (7) Patients underwent emergent surgeries, reoperations, and combined surgeries. In total, 88 T2DM patients were enrolled, and 8 of them received the intra-aortic balloon pump (IABP) during operation. Off-pump CABG was performed using conventional techniques, and complete revascularization was achieved in all patients. The mean duration of operation was around

4 h. Preoperative urine samples collected on the day before operation were named A group. At about 24 h after operation, postoperative urine samples were collected from a Foley catheter in the ICU and named B group. According to whether AKI occurred after operation, patients were grouped in the AKI group (AKI, $n = 44$) or the non-AKI group (NAKI, $n = 44$). Every patient in each group supplied both pre-and postoperative urine samples; thus, the AKI group was further subdivided into the AKI-A group and AKI-B group. The NAKI group was subdivided into the NAKI-A group and NAKI-B group. There were no deaths or dialysis needed in the AKI or NAKI groups. Serum creatinine of patients in the AKI group decreased to baseline when discharged.

Urine samples were also collected from early DN (DN-micro, $n = 46$) and overt DN (DN-macro, $n = 34$) patients as the validation cohort between Feb 2019 and Aug 2019 in our hospital. The inclusion criteria were as follows: (1) Age ranged from 30 to 75 years; (2) eGFR >30 mL/min/1.73 $m^2$; (3) Duration of T2DM >5 years. Exclusion criteria were as follows: (1) Other etiology of CKD; (2) AKI within three months; (3) Acute and chronic inflammation of the urinary system or other systems; (4) Patients with significant coexisting illnesses likely to affect survival or combined with other systemic diseases; (5) Patients with insufficient clinical data. The variables recorded for each participant included gender, age, body mass index (BMI), systolic blood pressure (SBP), diastolic blood pressure (DBP), diabetes duration, medical history, hemoglobin, albumin, fasting blood glucose (FBG), hemoglobin A1c (HbA1c), low-density lipoprotein (LDL), high-density lipoprotein (HDL), total cholesterol (CHOL), triglyceride (TG), serum creatinine (Scr), uric acid (UA), blood urea nitrogen (BUN), urinary NGAL, and UAER.

All urine samples were collected in sterile tubes (Corning Incorporated, Corning, NY, USA) and placed immediately on ice. Cell debris of samples was removed by centrifugation (2,000 rpm at 4 °C for 12 min). The supernatant was frozen at −80 °C for storage and subsequent extraction and analysis. All methods used in this study were approved by the Ethics Committee of the Second Xiangya Hospital, Central South University (IRB2017-S551). Written informed consent was collected from each patient before the study.

## Chemicals

Methyl alcohol, acetonitrile, and ethyl alcohol were purchased from Merck Company, Germany. Milli-Q system (Millipore Corp., Bedford, MA, USA) ultrapure water was used throughout the study. Authentic standards were purchased from Sigma-Aldrich (St. Louis, MO, USA). Two standards of metabolites (trans-4-hydroxy-L-proline and azelaic acid) were purchased from Sigma-Aldrich. All chemicals were of analytical grade.

## Urine preparation for metabolite extraction

The freeze-dried samples were crushed using a mixer mill (MM 400, Retsch) with a zirconia bead for 1.5 min at 30 Hz. Then, 100 mg of powder was extracted with 1.0 mL 70% aqueous methanol containing 0.1 mg/L lidocaine overnight at 4 °C. After 10,000 g centrifugation for 10 min, the supernatant was collected and filtered (SCAA-104, 0.22-μm pore size; ANPEL, Shanghai, China) prior to liquid chromatography–tandem mass spectrometry (LC–MS/MS) analysis. Quality Control (QC) samples were mixed by all samples to detect reproducibility of the whole experiment.

## Untargeted metabolomics analysis of the pilot study

The extracted preoperative and postoperative urine samples from 12 patients were used in the pilot study. An LC-ESI-MS/MS system (UPLC, Shim-pack UFLC SHIMADZU CBM30A; MS/MS, Applied Biosystems 6500 QTRAP) were used to analyze the extracted compounds (*Chen et al., 2013b*). For analysis, 2- µL samples were injected onto a Waters ACQUITY UPLC HSS T3 C18 column (2.1 mm*100 mm, 1.8 µm) operated at 40 °C. The mobile phases used were 0.04% acidified water (Phase A) and 0.04% acidified acetonitrile (Phase B). Through the following gradient program process, compounds were effectively separated: 5% B at 0 min; 95% B at 11.0 min; 95% B at 12.0 min; 5% B at 12.1 min; and 5% B at 15.0 min. The flow rate was 0.4 mL/min. The parameters used in the analysis are as follows: 500 °C electrospray ionization (ESI) source temperature; 5500 V ion spray voltage (IS); 25 psi curtain gas (CUR); the collision-activated dissociation (CAD) was set the highest. QQQ scans were acquired as multiple reaction monitoring (MRM) experiments with optimized declustering potential (DP) and collision energy (CE) for each individual MRM transition. The m/z range was set between 50 and 1000.

Data filtering, peak detection, alignment, and calculations were performed using Analyst 1.6.1 software (AB Sciex). Signal/noise >10 peaks were checked manually. In-house software written in Perl was used to remove redundant signals. Accurate m/z for each Q1 was acquired to identify the metabolites. Total ion chromatograms (TICs) and extracted ion chromatograms (EICs or XICs) of QC samples were used to construct metabolite profiles of all samples. MassBank, KNApSAcK, HMDB (*Wishart et al., 2013*), MoTo DB, and METLIN (*Zhu et al., 2013*) were the internal databases and public databases used to identify metabolites. The m/z values, RT, and fragmentation patterns of metabolites were compared with standards.

## Targeted metabolomics analysis of the validation study

Urine samples collected from 38 NAKI patients (24 h after operation), 38 AKI patients (24 h after operation), 46 early DN patients, and 34 overt DN patients were used in the validation study. Each urine sample was thawed on ice. Methanol was added to urine sample (3:1, v/v), then vortex-mixed for 30 s and centrifuged at 12,000 rpm for 15 min at 4 °C. The supernatant was collected and placed at −40 °C for 1 h. After centrifugation (12,000 rpm for 15 min at 4 °C), supernatant was placed in the vial for analysis. Standard stock solutions of trans-4-hydroxy-L-proline and azelaic acid were prepared in water/methanol (1:1, v/v) at the concentration of 1 mg/mL. A series of trans-4-hydroxy-L-proline standards was prepared at the following concentration levels:10, 20, 50, 100, 200 and 500 ng/mL. A series of azelaic acid standards was prepared at the following concentration levels: 20, 50, 100, 200, 500, 1000, and 2000 ng/mL. Gradient diluted standards were used to establish standard curves.

Quantitative analysis was performed using an LC-ESI-MS/MS system (Waters Acquity UPLC; MS/MS: Applied Biosystems API 5500 QQQ-MS). 4- µL prepared samples were applied to the ACQUITY UPLC BEH T3 column (100*2.1 mm, 1.7 µm) at 40 °C. The mobile phases were composed of solvent A (water with 0.02% formic acid) and solvent B (acetonitrile), and the following gradient elution conditions were used with a flow rate of
0.35 mL/min: 5% B at 0 min; 10% B at 1.0 min; 70% B at 2.0 min; 90% B at 3 min; 90% B at 4.0 min; 10% B at 4.01 min; 10% B at 5.0 min. The following parameters were set for the mass spectrometer: temperature 550 °C, ion spray voltage 5500 V, CUR 35 arb, CAD 7 arb, ionsource GS1 55 arb, and ionsource GS2 55 arb. According to the above conditions, the standard solutions were sampled separately. Retention time of trans-4-hydroxy-L-proline was 0.68 min, and azelaic acid was 2.53 min. Standard curves were generated by plotting peak areas of the mass chromatogram (y) versus the concentration (x) and fitting to a linear regression (Trans-4-hydroxy-L-proline, $y = 136.32x-686.72$, R2=0.9997; azelaic acid, $y = 6888.3x+311038$, R2=0.9967). The identification of target metabolites in urine sample was based on the spectral pattern and retention time. Trans-4-hydroxy-L-proline and azelaic acid were quantified based on their peak areas and standard curves.

The methods were validated based on parameters specified for bioanalytical methods according to U.S. Food and Drug Administration document and other related guidelines (*Gonzalez et al., 2014*). Selectivity, limit of detection (LOD) and limit of quantification (LOQ), linearity, precision and accuracy were evaluated. The LOD was defined as the concentration at a signal-to-noise ratio (S/N) of $\geq 3$, and the LOQ was defined at S/N $\geq 10$ with precision lower than 20%. Each sample was analyzed in biological replicates of three and technical replicates of three in this experiment.

### Data analysis

SPSS 23.0 software (SPSS Inc., Chicago, IL, USA) and GraphPad Prism 6.0 software were used for statistical analysis. Continuous variables were expressed as the means $\pm$ SD and were compared using a Student's $t$ test, a Wilcoxon test, one-way ANOVA or receiver operating characteristic (ROC) curves analysis. Categorical data were expressed as percentages (%) where appropriate and analyzed using Chi-square tests or Fisher's exact test. Statistical significance was considered if two-sided $p < 0.05$. Multivariate principal component analysis (PCA), partial least squares discriminant analysis (PLS-DA), orthogonal projection to latent structures- discriminant analysis (OPLS-DA), loadings plot and $T$-test, and univariate ANOVA were carried out to determine the differential metabolites between groups. A variable importance in projection (VIP) value of OPLS-DA model was applied to rank the metabolites that best distinguished between groups. Metabolites between two groups were considered different when the $p$-value of the $T$-test <0.05 and the variable importance in projection (VIP) $\geq 1$. Metabolites were mapped to the Kyoto Encyclopedia of Genes and Genomes (KEGG) database to determine important metabolic pathways and enrichment analysis. The calculated $p$-value was validated through false discovery rate (FDR) correction, taking FDR $\leq 0.05$ as a threshold.

## RESULTS

### Participant characteristics

This study was carried out in two stages, as shown in Fig. 1. Clinical characteristics of all study patients are summarized in Table 1. A group of 88 T2DM inpatients, who underwent the elective off-pump CABG procedure, were enrolled in our study. The mean age of the study population was 59.15 $\pm$ 8.76 years, and 50% were male. The mean eGFR was

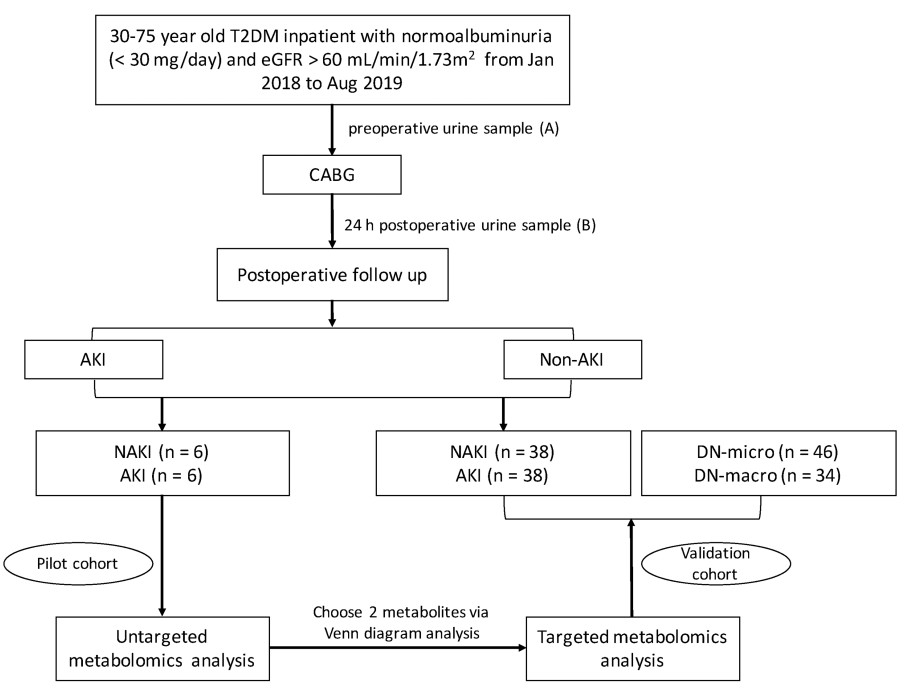

**Figure 1  Flow diagram of study design.** Abbreviations: CABG, coronary artery bypass grafting.

$92.28 \pm 17.67$ mL/min 1.73 m$^2$. The mean diabetes duration was $10.00 \pm 5.12$ years. Urine samples were collected from these patients before and 24 h after CABG. Among the 44 patients in the AKI group, 70.5% were in AKI stage 1, 22.7% were in AKI stage 2, and only 6.8% were in AKI stage 3. Six patients from the AKI group and six patients from the NAKI group were chosen as the pilot cohort. The preoperative clinical characteristics of two groups were matched. After operation, as expected, Scr and uNGAL in the AKI group were significantly higher than that in the NAKI group (Scr, $p = 0.001$; uNGAL, $p = 0.002$). The untargeted metabolomics analysis on the pilot cohort was aimed to identify biomarkers of postoperative AKI among T2DM patients. Two significantly altered metabolites found in untargeted metabolomics were quantified in the validation cohort by targeted metabolomics analysis. Postoperative urine samples from 38 NAKI patients and 38 AKI patients, urine samples from 46 early DN patients and 34 overt DN patients were used in the validation phase. Among the validation cohort, around 20% of T2DM patients receiving the CABG procedure had diabetic retinopathy, while more than 60% of DN patients had diabetic retinopathy. In addition, 26% early DN patients and 29.4% overt DN patients had a history of ischemic heart disease. Age, gender, diabetes duration, BMI, FBG, and HbA1c were not significantly different between the four groups. SBP, TG, UAER and number of patients with anemia were significantly higher in the DN-macro group, compared with the other three ($p < 0.001$). Renal function (BUN, Scr, eGFR) of the AKI group was the worst, followed by DN-macro, DN-micro, and NAKI groups ($p < 0.001$).

## Results of untargeted metabolomics in the pilot cohort

### The stability of mass signals and accuracy

Metabolomics showed very stable performance as TIC chromatograms from all QC samples were overlapping in both positive ion and negative ion models (Fig. S1). Each of the different colored peaks in positive ion (Fig. S2A) and negative ion models (Fig. S2B) represents one MRM metabolite detected in the urine specimens. As it shown in Fig. 2, the QC samples were tightly clustered in PCA scores plot, indicating high stability and reproducibility in the whole run. LC-MS/MS analysis of the discovery cohort urine samples identified a total of 742 known metabolites.

### Metabolic profiling of preoperative samples and postoperative samples

The metabolic profiling of the A group and B group was evaluated to explore the postoperative differential metabolites among DM. The PCA score plot indicated the separation between the A group and B group in the discovery set (Fig. 3A). The OPLS-DA model was constructed to obtain better discrimination between the above groups. As shown in Fig. 3B, each code represents a tested urine sample. The loading plot of OPLS-DA clearly separated tested samples into two blocks according to their metabolic profiles of different groups. These results suggest that the biochemical metabolites in urine changed significantly after the CABG procedure among DM patients. According to the $p$-value of the $T$-test, combined with VIP values of the OPLS-DA model, 61 metabolites were statistically distinguishable between the two groups (R2X = 0.407, R2Y = 0.908, Q2 = 0.836) and are shown in Supplementary Table In Fig. 3C, heat map analysis shows significantly different metabolites in the urine samples of the A and B groups, with 25 being presented at higher levels and 36 at lower levels in the B group compared with A group. In total, 22 pathways of KEGG classification were identified from 61 differential metabolites. KEGG pathway annotation showed that differentially expressed metabolites in DM patients after CABG were enriched in carbohydrate and amino acid metabolisms (Fig. 3D). The functional mechanism of selected metabolomics was based on the pathway enrichment analysis using KEGG. The top 20 pathways are shown in Fig. 3E. The top five metabolic pathways with the highest $p$-values in the metabolic pathway enrichment analysis were phenylalanine metabolism, galactose metabolism, citrate cycle, prolactin signaling pathway, and proximal tubule bicarbonate reclamation.

### Screening differential metabolites related to postoperative AKI in DM

We analyzed another two sets of pairwise data to identify AKI-related metabolites among DM patients: (i) AKI-B vs. AKI-A, (ii) AKI-B vs. NAKI-B. Two OPLS-DA models were constructed for these two comparisons. Figure 4A shows a clear separation between AKI-B and AKI-A. In total, 41 metabolites were differentially expressed in the AKI-B, compared with the AKI-A group; of these, 18 were up-regulated and 23 were down-regulated (Fig. 4B). Figure 4C shows a clear separation between AKI-B and NAKI-B. Moreover, 35 metabolites were differentially expressed in the AKI-B group, compared with the NAKI-B group. Of these, 23 were up-regulated and 12 were down-regulated (Fig. 4D).

The differential metabolites between AKI-B and AKI-A groups represent the postoperative AKI-related metabolites. Because there were 41 metabolites in these data sets,

Wang et al. (2020), *PeerJ*, DOI 10.7717/peerj.9111

**Table 1  Clinical characteristics of pilot and validation patients.**

| Variables | Pilot cohort | | | Validation cohort | | | | |
|---|---|---|---|---|---|---|---|---|
| | NAKI (*n* = 6) | AKI (*n* = 6) | *p* value | NAKI (*n* = 38) | AKI (*n* = 38) | DN-micro (*n* = 46) | DN-macro (*n* = 34) | *p* value |
| Age (years) | 59.00 ± 6.13 | 55.66 ± 7.39 | 0.415 | 60.28 ± 9.39 | 58.60 ± 8.73 | 58.13 ± 10.29 | 61.32 ± 8.35 | 0.406 |
| Male (%) | 50 | 50 | 1.000 | 47.4 | 52.6 | 52.2 | 58.8 | 0.822 |
| DM duration (years) | 10.33 ± 6.71 | 9.66 ± 6.28 | 0.863 | 10.21 ± 5.05 | 9.78 ± 4.96 | 12.34 ± 6.78 | 11.85 ± 5.53 | 0.192 |
| Medical history (%) | | | | | | | | |
| Diabetic retinopathy | 16.7 | 16.7 | 1.000 | 18.4 | 21.1 | 69.6 | 67.6 | <0.001 |
| Stroke | 0 | 0 | 1.000 | 5.3 | 7.9 | 8.7 | 11.8 | 0.799 |
| BMI | 22.73 ± 3.71 | 25.60 ± 4.28 | 0.244 | 24.19 ± 3.05 | 25.13 ± 2.74 | 24.55 ± 2.90 | 24.74 ± 2.46 | 0.529 |
| SBP (mm Hg) | 138.50 ± 30.98 | 146 ± 19.45 | 0.619 | 145.44 ± 21.82 | 147.18 ± 22.45 | 145.28 ± 20.29 | 164.64 ± 20.07[*,a,b] | <0.001 |
| DBP (mm Hg) | 77.16 ± 11.58 | 83.33 ± 6.94 | 0.290 | 80.34 ± 13.66 | 81.55 ± 8.68 | 83.76 ± 14.45 | 87.73 ± 15.54 | 0.102 |
| Serum albumin (g/L) | 42.06 ± 2.79 | 42.25 ± 2.51 | 0.907 | 42.30 ± 4.20 | 41.82 ± 5.00 | 37.01 ± 4.85[*,a] | 34.07 ± 5.57[*,a,b] | <0.001 |
| Anemia (%) | 16.7 | 16.7 | 1.000 | 13.2 | 10.5 | 19.6 | 35.3[*,a,b] | 0.038 |
| FBG (mmol/L) | 6.84 ± 1.23 | 7.14 ± 1.83 | 0.748 | 7.07 ± 2.97 | 7.52 ± 2.43 | 7.66 ± 2.15 | 7.61 ± 3.02 | 0.752 |
| HbA1c (%) | 7.83 ± 0.92 | 7.58 ± 1.01 | 0.664 | 7.75 ± 1.87 | 8.36 ± 2.22 | 8.89 ± 2.26 | 8.76 ± 2.10 | 0.082 |
| Scr (*μ*mol/L) | 76.65 ± 12.98 | 82.38 ± 18.08 | 0.542 | 76.02 ± 17.08[a] | 139.93 ± 46.16 | 78.71 ± 24.66[a] | 92.99 ± 22.30[a,b] | <0.001 |
| Postoperative Scr (*μ*mol/L) | 79.91 ± 10.61 | 145.11 ± 31.14 | 0.001 | | | | | |
| BUN (mmol/L) | 6.18 ± 1.06 | 7.46 ± 1.69 | 0.150 | 6.70 ± 1.59[a] | 10.05 ± 2.74 | 6.79 ± 2.16[a] | 7.64 ± 2.13[a] | <0.001 |
| Postoperative BUN (mmol/L) | 7.13 ± 2.55 | 10.01 ± 3.27 | 0.120 | | | | | |
| UA (*μ*mol/L) | 419.23 ± 138.82 | 374.45 ± 93.78 | 0.527 | 352.56 ± 87.74 | 358.85 ± 82.61 | 343.08 ± 90.85 | 304.36 ± 72.08[a] | 0.034 |
| Postoperative UA (*μ*mol/L) | 324.58 ± 173.92 | 375.00 ± 84.70 | 0.538 | | | | | |
| eGFR (mL/min 1.73 m²) | 88.01 ± 24.39 | 85.80 ± 25.30 | 0.880 | 83.09 ± 16.60[a] | 46.68 ± 16.54 | 86.14 ± 25.00[a] | 72.91 ± 19.36[a,b] | <0.001 |
| Postoperative eGFR (mL/min 1.73m²) | 84.45 ± 26.27 | 43.80 ± 18.27 | 0.011 | | | | | |
| Postoperative urinary NGAL (*μ*mol/L) | 15.75 ± 14.55 | 324.39 ± 134.46 | 0.002 | 12.84 ± 15.44 | 303.74 ± 120.85 | | | <0.001 |
| TG (mmol/L) | 1.71 ± 0.53 | 1.37 ± 0.57 | 0.302 | 1.74 ± 0.65 | 2.27 ± 1.00 | 2.10 ± 0.93 | 3.16 ± 2.40[*,a,b] | <0.001 |
| CHOL (mmol/L) | 4.18 ± 1.19 | 3.95 ± 0.64 | 0.686 | 4.29 ± 1.28 | 5.03 ± 1.30 | 4.16 ± 1.20[a] | 5.15 ± 1.43[*,b] | 0.001 |
| HDL (mmol/L) | 0.93 ± 0.09 | 1.10 ± 0.16 | 0.063 | 0.99 ± 0.25 | 1.05 ± 0.32 | 0.99 ± 0.21 | 1.09 ± 0.25 | 0.238 |
| LDL (mmol/L) | 2.85 ± 1.04 | 2.40 ± 0.60 | 0.383 | 2.79 ± 1.10 | 3.24 ± 1.01 | 2.62 ± 1.15 | 3.35 ± 1.31[b] | 0.012 |
| UAER (mg/day) | 7.58 ± 8.01 | 8.13 ± 9.96 | 0.918 | 13.57 ± 9.11 | 12.41 ± 9.07 | 88.45 ± 51.27 | 1533.63 ± 1096.98[*,a,b] | <0.001 |

**Notes.**

Data are presented as N (%) or mean ± SD. *p* < 0.05 was considered significant.

[*]*p* < 0.05 vs. NAKI group.

[a]*p* < 0.05 vs. AKI group.

[b]*p* < 0.05 vs. DN-micro group.

Abbreviations: BMI, body mass index; SBP, systolic blood pressure; DBP, diastolic blood pressure; FBG, fasting blood glucose; HbA1c, haemoglobin A1c; LDL, low-density lipoprotein; HDL, high-density lipoprotein; CHOL, total cholesterol; TG, triglyceride; eGFR, estimated glomerular filtration rate; Scr, serum creatinine; UA, uric acid; BUN, blood urea nitrogen; UAER, urinary albumin excretion rate.
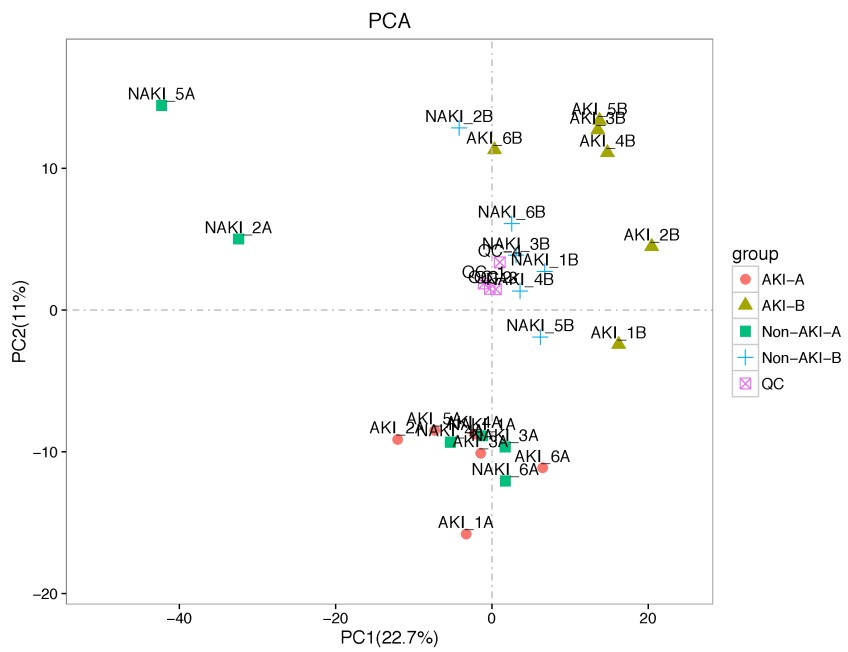

**Figure 2** **PCA performed on all samples.** Urine samples in different groups are represented by differently colored symbols. Abbreviations: PCA, principal components analysis.

it was particularly important to select the metabolites with the greatest association with postoperative AKI among DM patients. The data of B vs. A provided us with information about operation-related differential metabolites. The data of AKI-B vs. NAKI-B provided us with information about AKI-related differential metabolites. Through Venn analysis, the postoperative AKI-related metabolites (the data of AKI-B vs. AKI-A) were compared with operation-related metabolites (the data of B vs. A) and AKI-related metabolites (the data of AKI-B vs. NAKI-B) to further reduce the number of postoperative AKI-related metabolites. Based on the results of Venn analysis, the nine most significant markers of postoperative AKI were found. The results show that the five up-regulated metabolites included trans-4-hydroxy-L-proline, uridine triphosphate, p-aminobenzoate, caffeic acid, and adrenochrome. The relative intensities of these metabolites in the A and B groups are shown in Figs. 5A–5F. The four down-regulated metabolites included δ-valerolactam, L-norleucine, 5′-deoxy-5′-(methylthio) adenosine, and azelaic acid. The relative intensities of these metabolites in the A and B groups are shown in Figs. 5G–5K. These nine overlapping metabolites shown in Table 2 were mainly distributed in the functional categories of amino acid, nucleotide, and organic acid metabolomics.

## Validation of potential markers

The discovered metabolite markers trans-4-hydroxy-L-proline and azelaic acid, which have the maximum absolute fold change value among the nine overlapping metabolites, were further quantified in the validation cohort by targeted metabolomics analysis. To better understand the possible role of these two metabolites in the acute and chronic development
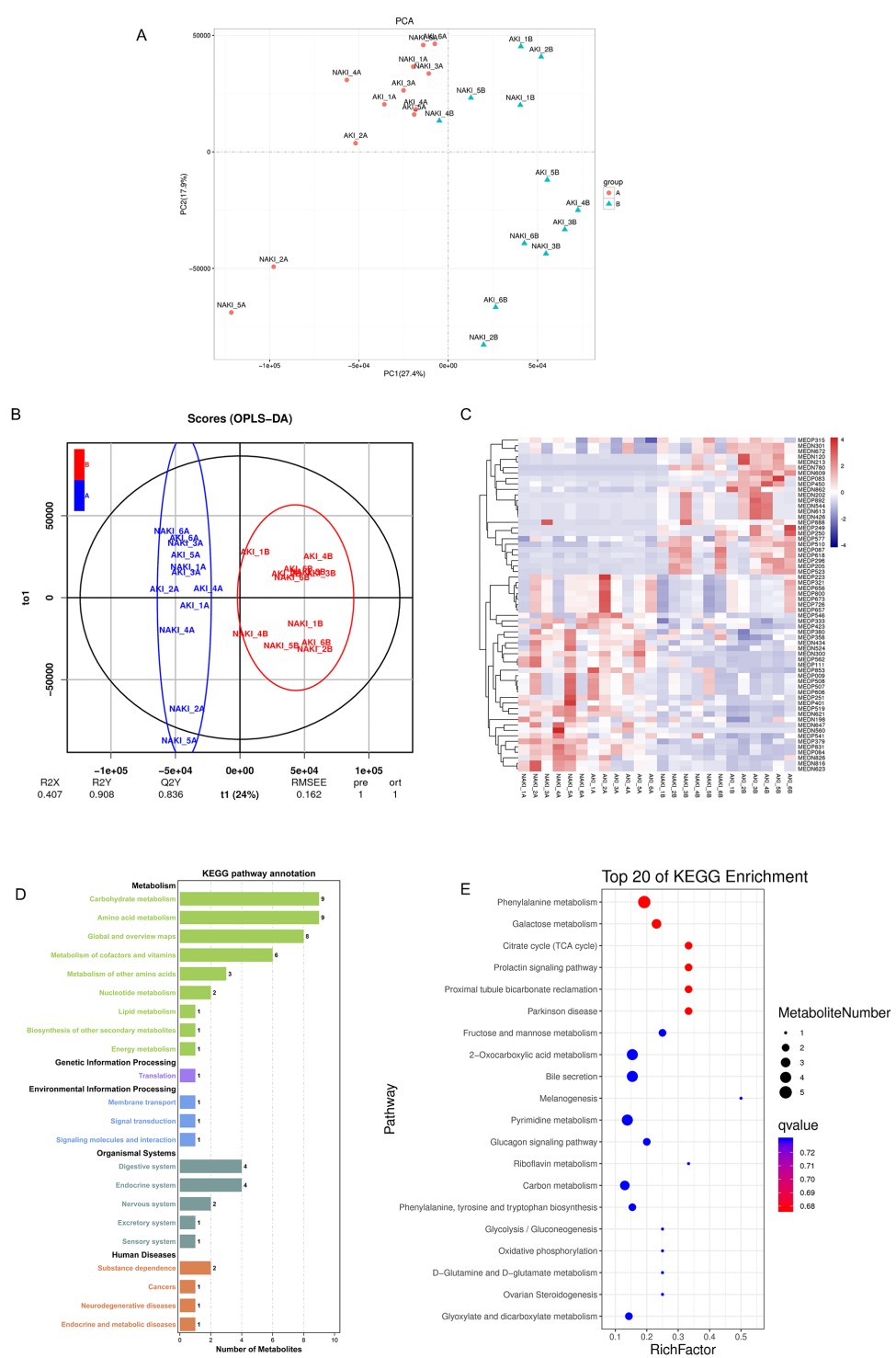

**Figure 3** **Separating postoperative samples from preoperative samples by metabolic profiling analysis.**
(A) Score scatter plot of PCA model for B group vs. A group; (B) Score scatter plot of OPLS-DA model for B group vs. A group; (C) Heat map representing statistically different (continued on next page…)

**Figure 3 (…continued)**
metabolites among two groups. Scale and metabolites category are provided to right of heat map; (D) KEGG pathways annotation based on significant different metabolites between B vs. A group. The $x$ axis indicates the proportion and number of metabolites annotated to the pathway, and the $y$ axis indicates name of the KEGG metabolic pathway; (E) Statistics of KEGG enrichment. The $x$ axis indicates the rich factor corresponding to each pathway, and the $y$ axis indicates name of the KEGG metabolic pathway. The size of bubble indicates number. The color of the point represents the p-values of the enrichment analysis. Abbreviations: PCA, principal components analysis; OPLS-DA, orthogonal projection to latent structures-discriminant analysis; A, preoperative sample; B, postoperative sample; KEGG, the Kyoto Encyclopedia of Genes and Genomes.

of renal injury among T2DM patients, NAKI ($n = 38$), AKI ($n = 38$), DN-micro ($n = 46$), and DN-macro ($n = 34$) were chosen as the validation group. The results are shown in Fig. 6. The absolute trans-4-hydroxy-L-proline concentration increased gradually from the NAKI group to the AKI, DN-micro and DN-macro groups ($p < 0.0001$) (Fig. 6A). On the basis of quantitative analysis results, ROC curves were applied to investigate the clinical diagnostic potentials of these metabolites. An AUC value approaching 1.0 indicates a better diagnostic effectiveness. As shown in Figs. 6B–6C, the AUC of trans-4-hydroxy-L-proline between AKI and DN-micro groups was 0.722 (Fig. 6B, $P = 0.0005$, 95% CI [0.612–0.833]). The AUC of trans-4-hydroxy-L-proline between AKI and DN-macro groups was 0.783 (Fig. 6C, $P < 0.0001$, 95% CI [0.669–0.898]). These data suggest that the abundances of urinary trans-4-hydroxy-L-proline increased from acute renal damage to chronic renal damage in DM. For azelaic acid, the highest level was found in the NAKI and DN-micro groups, followed by the DN-macro group. The lowest level was found in the AKI group ($p < 0.0001$, Fig. 6D). The azelaic acid concentrations in the validation cohort were positively correlated with eGFR levels ($p < 0.0001$), and the Pearson correlation coefficient was 0.539 (Fig. 6E). As shown in Figs. 6F–6G, the AUC of azelaic acid between NAKI and AKI groups was 0.860 (Fig. 6F, $P < 0.0001$, 95% CI [0.770–0.949]). The AUC of azelaic acid between DN-micro and DN-macro groups was 0.754 (Fig. 6G, $P = 0.0001$, 95% CI [0.648–0.860]). These data suggest that the abundances of urinary azelaic acid decreased with acute or chronic deterioration of renal function in DM.

## DISCUSSION

Many studies reported that DM increases the risk of CABG postoperative complication (*Brush Jr et al., 2019*; *Hallberg et al., 2014*; *Shafranskaya et al., 2015*). However, this is the first study to use untargeted metabolomics to exam metabolic changes of DM patients who experience operation-like acute attack. Via screening these operation-related metabolites, we identified some novel postoperative AKI markers of DM, and we further proved their role in the chronic progression of DN via targeted metabolomic analysis.

A prospective nested case-control study reported that after cardiopulmonary bypass surgery, all patients (with and without AKI) experienced tubular injury and stress, evidenced by proteinuria and the appearance of 2-microglobulinuria (*Ho et al., 2009*). In order to study the general metabolic changes of DM patients after CABG surgery, we enrolled T2DM inpatients who underwent the CABG procedure in our hospital. Six patients from the AKI group and six patients from the NAKI group were chosen as the pilot cohort.

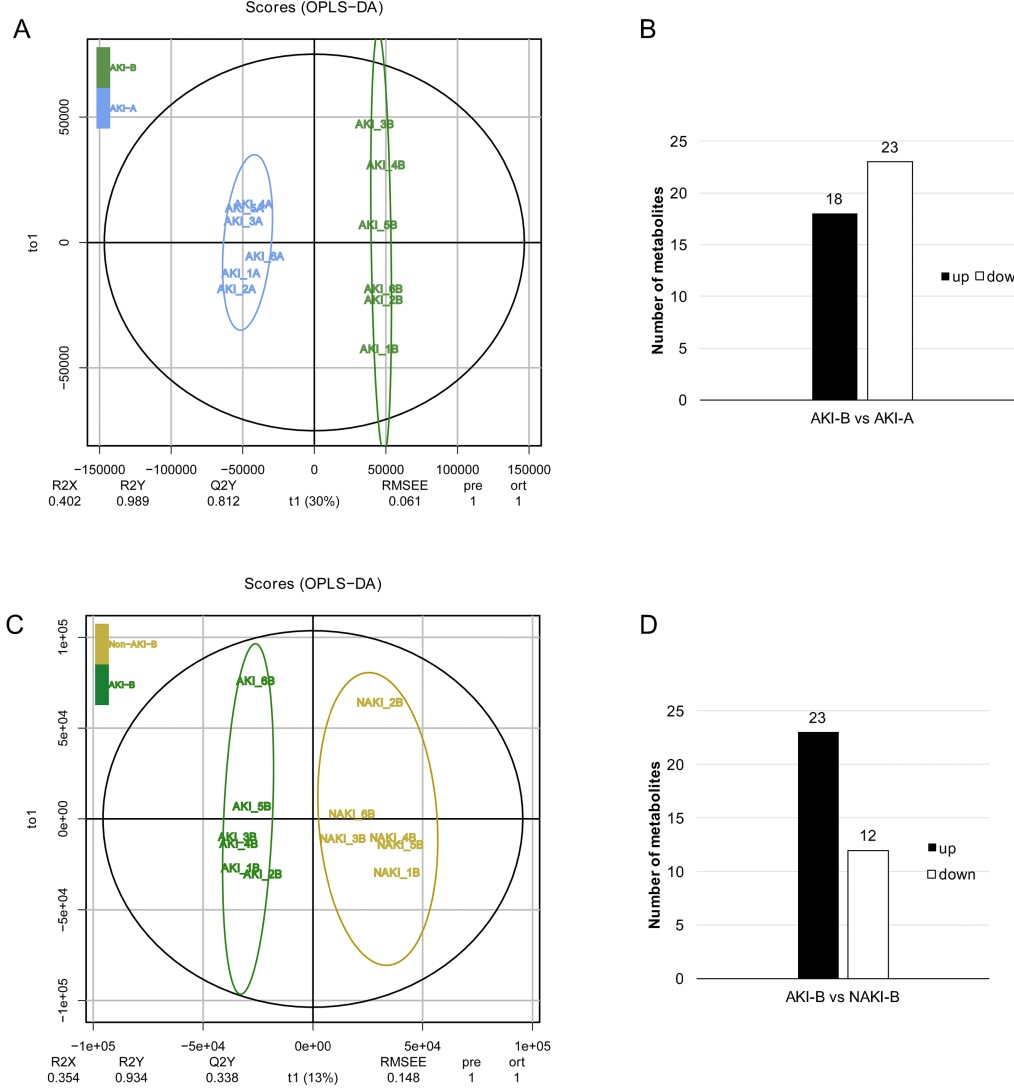

**Figure 4** **OPLS-DA score plot of of AKI-B vs. AKI-A and AKI-B vs. NAKI-B.** (A, B) OPLS-DA score plot of AKI-B vs. AKI-A, eighteen and twenty-three up- and down-regulated metabolites, respectively, in the AKI-B group compared with the AKI-A group; (C, D) OPLS-DA score plot of AKI-B vs. NAKI-B, twenty-three and twelve up- and down-regulated metabolites, respectively, in the AKI-B group compared with the NAKI-B group. Abbreviations: OPLS-DA, orthogonal projection to latent structures- discriminant analysis; A, preoperative sample; B, postoperative sample.

Via untargeted metabolomics analysis, we found the expression of 61 urine metabolites changed significantly after operation in all 12 patients (with and without AKI). Of interest, these operation-related metabolites in DM patients, which represent activated metabolic pathways in acute phase after operation, have been associated with chronic DN progression in previous reports.

We found urinary hippuric acid was elevated after operation, and indoxylsulfuric acid decreased after operation. Hippuric acid is converted from dietetic aromatic compounds by

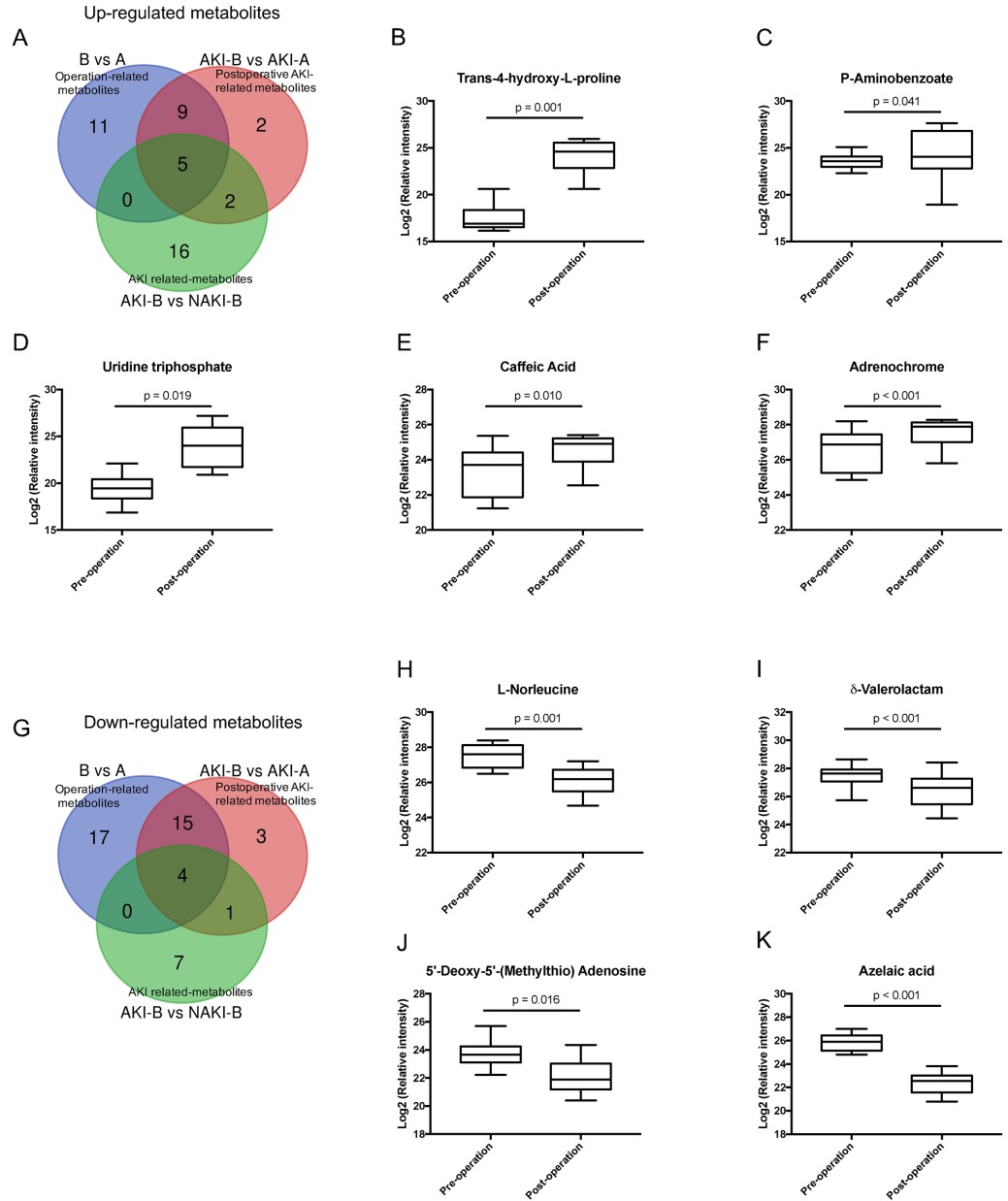

**Figure 5  Venn analysis identified metabolites associated with postoperative AKI in DM.** (A) Venn diagram of the up-regulated overlapping metabolites from three pairs data (B vs. A, AKI-B vs. AKI-A and AKI-B vs. NAKI-B); (B–F) The relative intensities of the five up-regulated overlapping metabolites before and after CABG; (G) Venn diagram of the down-regulated overlapping metabolites from three pairs data (B vs. A, AKI-B vs. AKI-A and AKI-B vs. NAKI-B); (H–K) The relative intensities of the four down-regulated overlapping metabolites before and after CABG. Abbreviations: CABG, coronary artery bypass grafting; A, preoperative sample; B, postoperative sample.

the gut microbiome, and high urinary hippuric acid had been linked to impaired glomerular filtration and tubular reabsorption (*Liu et al., 2013*). DN patients showed increased levels of urinary hippuric acid, compared with the healthy and T2DM patients (*Li et al., 2017*).

**Table 2  The nine urine overlapping metabolites potentially associated with postoperative AKI in DM.**

| Metabolite | Change trend | B vs. A | | | AKI-B vs. AKI-A | | | AKI-B vs. NAKI-B | | |
|---|---|---|---|---|---|---|---|---|---|---|
| | | log2 fold change | *p* value | VIP | log2 fold change | *p* value | VIP | log2 fold change | *p* value | VIP |
| Trans-4-hydroxy-L-proline | up | 6.426 | 0.001 | 1.991 | 6.912 | 0.005 | 2.376 | 3.513 | 0.003 | 5.468 |
| Uridine triphosphate | up | 5.117 | 0.019 | 1.828 | 6.625 | 0.029 | 2.612 | 1.579 | 0.017 | 2.628 |
| P-Aminobenzoate | up | 2.159 | 0.041 | 1.982 | 3.487 | 0.010 | 3.618 | 2.704 | 0.030 | 3.415 |
| Caffeic Acid | up | 0.966 | 0.010 | 1.049 | 1.366 | 0.013 | 1.608 | 0.923 | 0.014 | 1.613 |
| Adrenochrome | up | 0.791 | <0.001 | 2.624 | 1.203 | 0.013 | 4.182 | 0.765 | 0.024 | 3.760 |
| $\delta$-Valerolactam | down | −0.744 | <0.001 | 2.819 | −1.149 | 0.004 | 2.429 | −1.455 | 0.032 | 5.574 |
| L-Norleucine | down | −1.330 | 0.001 | 3.745 | −1.886 | 0.018 | 4.461 | −1.078 | 0.006 | 3.888 |
| 5′-Deoxy-5′-(Methylthio) Adenosine | down | −1.519 | 0.016 | 1.057 | −1.938 | 0.020 | 1.065 | −1.244 | 0.008 | 1.223 |
| Azelaic acid | down | −3.338 | <0.001 | 2.768 | −4.254 | 0.007 | 3.110 | −1.674 | 0.048 | 1.161 |

**Notes.**

Metabolites between two groups were considered different when the *p*-value of the *T*-test <0.05 and VIP ≥ 1.

Abbreviations: A, preoperative sample; B, postoperative sample; VIP, variable importance in projection.

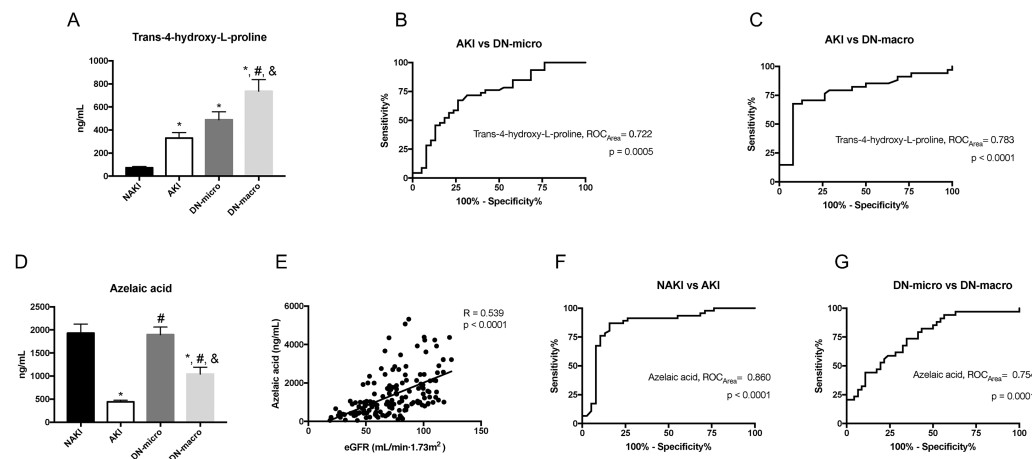

**Figure 6  Urine metabolomic abundance in validation cohort were measured by targeted metabolomics analysis.** (A) The absolute concentration levels of trans-4-hydroxy-L-proline in NAKI, AKI, DN-micro and DN-macro groups; (B–C) ROC curves of trans-4-hydroxy-L-proline for distinguishing AKI from DN-micro and DN-macro patients; The sensitivity (true positive rate) is set as the ordinate, and 1-specificity (false positive rate) is set as the abscissa. (D) The absolute concentration levels of azelaic acid in NAKI, AKI, DN-micro and DN-macro groups. (E) Correlation of azelaic acid with eGFR in cohort patients. Pearson correlation coefficient (R) is provided in the picture (F–G) ROC curves of azelaic acid for distinguishing NAKI from AKI patients, DN-micro from DN-macro patients. $p < 0.05$ was considered significant. *, $p < 0.05$ vs. NAKI group; #, $p < 0.05$ vs. AKI group; &, $p < 0.05$ vs DN-micro group. Abbreviation: ROC, receiver operating characteristic curves.

Indoxylsulfuric acid is one of the uremic toxins derived from dietary proteins. In DM patients and animal models of DN, serum indoxylsulfuric acid levels increased with disease progression, and this was accompanied by the progression of renal function impairment (*Atoh, Itoh & Haneda, 2009*).

The tricarboxylic acid (TCA) cycle coordinates glucose, fatty acid, and amino acid metabolism. Key TCA cycle metabolites include lactate, pyruvate, citric acid, a-ketoglutaric acid, succinate, fumarate, and malate. We found after the CABG procedure, urinary citric acid significantly increased, whereas a-ketoglutaric acid significantly decreased. Elevated urinary TCA cycle metabolites have been observed in db/db mice previously. Changes in urinary TCA cycle intermediates were also reported in DN participants who experienced progressive CKD, compared with those with stable renal function (*Li et al., 2013*; *Liu et al., 2018*).

Urinary aromatic AA-tyrosine decreased after the CABG procedure in DM patients. Changes in urine tyrosine have been reported in T2DM patients with micro-macro albuminuria, compared with healthy and DM patients (*Chen et al., 2018a*, *Pena et al., 2014*). Besides the metabolites mentioned above, trimethylamine N-oxide, azelaic acid and sugars, including D-sorbitol and D-glucose, also changed after the operation in our study. These changes have been reported in association with chronic DN progression in other studies (*Chen et al., 2013a*; *Hirayama et al., 2012*; *Klein & Shearer, 2016*; *Selby, Shearing & Marshall, 1995*; *Winther et al., 2019*). In our study, some of our operation-related metabolites overlapped with some chronic DN progression metabolites reported in other studies. This indicated that, for DM patients, the pathological changes after acute events, such as an operation, can be detected by metabolomics and may be involved in the chronic progression of DN.

DM is a known risk factor of AKI after cardiovascular surgery (*Hertzberg, Sartipy & Holzmann, 2015*). What effect does AKI occurring in DM will exert to kidney? The ischemic AKI induces significant interstitial inflammation and functional impairment/structural damage of small peritubular and glomerular blood vessels. Ischemic AKI in DM can further aggravate renal chronic inflammation and micovascular damage, which may affect the kidney in the long-term (*He et al., 2017*; *Kelly, Burford & Dominguez, 2009*; *Patschan & Muller, 2016*). Via non-invasive metabolomics of urine samples, we have realized the dynamic observation of short-term and long-term effects of AKI occurring in DM on kidney. After screening operation-related metabolites, we found 9 of 61 were postoperative AKI-related metabolites, including trans-4-hydroxy-L-proline, uridine triphosphate, p-aminobenzoate, caffeic acid, adrenochrome, δ-valerolactam, L-norleucine, 5′-deoxy-5′-(methylthio) adenosine, and azelaic acid. Expression of these metabolites represents a single episode of kidney injury happen on DM. We speculate that each episode of kidney injury has a cumulative dose–response association. Several episodes may gradually accumulate and partially contribute to chronic DN progression. Among the nine metabolites, we choose two metabolites with the maximum absolute fold change value, trans-4-hydroxy-L-proline and azelaic acid, to verify our speculation by the validation cohort.

Trans-4-hydroxy-L-proline is one isomeric form of hydroxyproline, which is mainly identified in collagen, and it was found to be a good indicator of collagen metabolism in bone disease. Trans-4-hydroxy-L-proline is an important constituent of collagen and is a collagen-specific amino acid, with approximately 1% of the amino acid found in elastin (*Srivastava et al., 2016*). Its level in circulation has been correlated with the degree of fibrosis in chronic hepatitis (*Attallah et al., 2007*; *Lawrence et al., 2019*). Moreover, abnormal

plasma trans-4-hydroxy-L-proline concentration was found in an insulin-dysregulated animal model (*Kenez et al., 2018*). Changes in trans-4-hydroxy-L-proline in circulation have been reported, along with collagen metabolism, fibrosis, and insulin-dysregulation. These changes seem to be related to DN progression, but no study has validated the contribution of trans-4-hydroxy-L-proline to early DN progression. In the validation cohort, we found urinary trans-4-hydroxy-L-proline concentration gradually increased with chronic DN progression, along with AKI episode. It is reported that every ischemic AKI insult diminishes the intrarenal total vascular surface area, subsequently followed or accompanied by endothelial-to-mesenchymal transdifferentiation (EndoMT) (*O'Riordan et al., 2007*; *Zeisberg et al., 2008*). EndoMT and epithelial-mesenchymal transition (EMT) was associated with increased collagen production and renal fibrosis (*Cruz-Solbes & Youker, 2017*; *Zeng, Xiao & Sun, 2019*). As an indicator of collagen metabolism, we speculated that the increase in urinary trans-4-hydroxy-L-proline after AKI may reflect mild renal fibrosis, which may be undetectable by normal assay and could be treated as an early warning of DN progression. With DN progression, AKI-like episodes and renal fibrosis increased, presenting as an accumulation of urinary trans-4-hydroxy-L-proline concentration.

Azelaic acid is a saturated C9 dicarboxylic acid derived from the oxidation of fatty acids. Azelaic acid inhibits the generation of reactive oxygen species on neutrophils (*Akamatsu et al., 1991*). As a medicine, it possesses radical scavenging (*Passi et al., 1991*), antimicrobial (*Charnock, Brudeli & Klaveness, 2004*), and antitumor (*Breathnach, 1999*) potentials. Azelaic acid was found to be effective in regulating high fat diet-induced oxidative stress and treating insulin resistance associated with T2DM in the mice model (*Muthulakshmi, Chakrabarti & Mukherjee, 2015*). As a metabolite, azelaic acid was used to distinguish patients with depression and anxiety disorders from healthy controls (*Chen et al., 2018b*). Compared with healthy individuals, patients with psoriasis had lower azelaic acid levels (*Kang et al., 2017*). The levels of azelaic acid in serum were lower in DN patients, compared with non-DN patients (*Hirayama et al., 2012*). Our study found that the level of urinary azelaic acid cannot associate the acute renal events of DM with the chronic progression of DN, but it has a negative correlation with both acute and chronic deterioration of renal function in DM. Thus, urinary azelaic acid could be considered used to monitor renal function noninvasively among DM and DN patients.

We must consider the limitations of our study. First, the study was conducted in a single-center. The findings should be validated in a larger multicenter cohort. Second, only two of the nine metabolites were confirmed in the validation cohort. The other seven metabolites need future study. Third, association but not causal links can be described between trans-4-hydroxy-L-proline, azelaic acid, and DN progression. The precise molecular mechanisms underlying the results are still unknown. Mechanistic studies are needed to clarify the exact role of these metabolites in DN progression. Lastly, our results are limited to micro- and macro-albuminuria DN, and it is necessary to test if our results apply in other conditions, such as the progression of DN to ESRD.

## CONCLUSION

Our study explored the effect of AKI in DM on chronic DN progression and visualized this process via metabolomics. For DM patients, the pathological changes after acute events, such as operation, can be detected by metabolomics. The effect of the postoperative AKI on DM could be measured via, but not limited to, trans-4-hydroxy-L-proline and azelaic acid. The influence of AKI might accumulate and partially contribute to the chronic progression of DN. The detection of urinary trans-4-hydroxy-L-proline after AKI could be treated as an early warning of DN progression and might be linked to renal fibrosis. Meanwhile, urinary azelaic acid was identified as a noninvasive renal function indicator among DM and DN patients. Our results not only identified markers of AKI on DM and the chronic progression of DN but also interpreted the progression of DN from a new perspective. We associated chronic DN progression with AKI occurring in DM, which might enable early monitoring and initiation of specific therapies in patients with DM or DN.

### Funding

This work was supported by the National Natural Science Foundation of China (81870500, 81770714 and 81570622). The funders had no role in study design, data collection and analysis, decision to publish, or preparation of the manuscript.

### Grant Disclosures

The following grant information was disclosed by the authors:
National Natural Science Foundation of China: 81870500, 81770714, 81570622.

### Competing Interests

The authors declare there are no competing interests.

### Author Contributions

- Jiayi Wang conceived and designed the experiments, performed the experiments, analyzed the data, prepared figures and/or tables, and approved the final draft.
- Wenzhe Yan and Xiang Zhou performed the experiments, prepared figures and/or tables, and approved the final draft.
- Yu Liu and Chengyuan Tang analyzed the data, prepared figures and/or tables, and approved the final draft.
- Youming Peng, Hong Liu, Lin Sun and Liyu He conceived and designed the experiments, authored or reviewed drafts of the paper, and approved the final draft.
- Li Xiao analyzed the data, authored or reviewed drafts of the paper, and approved the final draft.

### Human Ethics

The following information was supplied relating to ethical approvals (i.e., approving body and any reference numbers):

The Medical Ethics Committee of the Second Xiangya Hospital, Central South University granted the ethical approval to carry out the study within its facilities (IRB2017-S551).

## Data Availability

The raw data are available in the Supplemental File.

## Supplemental Information

Supplemental information for this article can be found online at http://dx.doi.org/10.7717/peerj.9111#supplemental-information.

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
