# Peer review of "Metabolomics window into the role of acute kidney injury after coronary artery bypass grafting in diabetic nephropathy progression"

_PeerJ, doi:10.7717/peerj.9111_

## Round 0.1 · original submission · Major Revisions

As suggested by the reviewers, the authors should enrich the introduction, add information about clinical data, improve the figures shown.

Reviewer 1 ·

Basic reporting

• There are some typing errors, so I would appreciate if the authors could check the entire manuscript through the text. Here are some examples:
-line 35: in the Abstract, please specify Diabetes Mellitus before using DM
-line 41: in the Abstract, “paitents” instead of “patients”
-line 71: “for perdicting development of…” instead of “for predicting development of…”
-line 90: “progrocess” instead of “progress”
-line 199: “metabolicpathways” instead of “metabolic pathways”
-line 321: “Key TCA cycle metabolites includes…” instead of “Key TCA cycle metabolites include…”
-line 327: “AA-tyrosine was decrease….” instead of “AA-tyrosine was decreased….”
-line 331: “reported in associations with….” instead of “reported in association with….”
-line 352: “one isomeric forms….” instead of “one isomeric form….”
……….

• As regard the Introduction, I think the authors should enrich that section, since there are interesting manuscripts to refer to and to comment on. As an example, I may suggest to mention and to discuss on the evidences highlighted in the manuscript by Wang TJ et. al., (Metabolite profiles and the risk of developing diabetes. Nat Med (2011)), by Zhang J et. al., (Metabonomics research of diabetic nephropathy and type 2 diabetes mellitus based on UPLCoaTOF-MS system. Anal Chim Acta (2009)), as well as the findings of “Proteomic and metabolomic characterization of diabetic nephropathy” (Acta Diabetologica (2018)) by Rossi C et.al.. I am sure comments on already described metabolomics characterization of DM and DN would better demonstrate the potential of metabolomics strategy by mass spectrometry in the study of such disorders.

• Figure 3 is difficult to appreciate in term of resolution, in particular Panel d and e, referring to the KEGG metabolic pathway and statistics of KEGG enrichment are both impossible to read.

• Figure 4, the colour code of each Panel are confusing: it seems that blue means A group and orange B group, but in the other Panels blue and orange code are used for up- and down-, and for AKI and NAKI. Thus, a reorganization of Figure 4 is strongly recommended.

• Also Figure 5 is not clear. Firstly, it is difficult to understand that the first 5 scatter plot in panel c are referred to Venn diagram in Panel a, and the last 4 scatter plot in panel c are referred to Venn diagram in Panel b. Moreover, significance information of t-test is missing in the comparison. It is also not clear why showing the third comparison (the green one) in the Venn diagram of panel a and b. Thus, I suggest a better explanation of the data processing.

Experimental design

• In the Materials and Methods section, in particular in “Targeted metabolomics analysis of the validation method”: it is really difficult for the reader to understand that this is a quantitative analysis. No analytical parameter, such as linearity, concentration range of interest, precision of the method, is described and no reference to a previously validated method is reported. This is a really important point.

Validity of the findings

Results, as shown in Figure 5 are not clear. Firstly, it is difficult to understand that the first 5 scatter plot in panel c are referred to Venn diagram in Panel a, and the last 4 scatter plot in panel c are referred to Venn diagram in Panel b. Moreover, significance information of t-test is missing in the comparison. It is also not clear why showing the third comparison (the green one) in the Venn diagram of panel a and b. Thus, I suggest a better explanation of the data processing in Figure and through the text.

Additional comments

I am writing about the manuscript of Jiayi Wang et. al. entitled “Metabolomics window into the role of postoperative acute kidney injury in chronic kidney injury among diabetes mellitus patients”.

Through an untargeted metabolomics strategy, metabolites statistically distinguishable were highlighted in postoperative urine samples compared to preoperative urine samples. Following, to better understand the role of identified AKI markers in chronic development of renal injury among T2DM patients, the over-highlighted metabolites were quantified by a targeted metabolomics approach.

Overall, this research article is well written and the aim of the manuscript is well described, bringing relevant observations on metabolomics approach as a valuable tool in the discovery of novel biomarkers of Diabetes Mellitus and Diabetic Nephropathy. Anyway, I have some comments and some aspects to be clarified that would be taken in consideration to improve the manuscript.

·

Basic reporting

Well-structured and designed manuscript/study, but following issues should be addressed.

#Title should include CABG, not just postoperative AKI.

Experimental design

#Why eGFR was calculated using MDRD, not CKD-EPI?
- over 60 ml/min/1.73m^2, CKD-EPI is accurate and recommended.

#Collection of urine: is there no emergent surgery? (preoperative samples in emergent CABG could not be used

#24h postoperative urine collection was done from immediate postoperative to 24 hours. Is this right? When the urine collection was started? Please describe the method in detail.

#How the author controlled the severity of AKI (e.g., staging of AKI)?
The severity of AKI could affect the relationships and findings.

Validity of the findings

#Please describe following information
- Information on CABG (IABP, CPB, surgery time, and so on)
- Information on diabetes (% of diabetic retinopathy)
- Information on others (anemia, albumin, Hx of ischemic heart disease and stroke)

#How was the recovery of AKI? The levels of metabolites were associated with recovery of AKI or progression to CKD (e.g., 3 month-Cr level)?

#Figures: P values should be described in figures (esp. Figure 5).

---

## Round 0.2 · Minor Revisions

The authors should fix the minor issues left (title and KDIGO guidelines, the most recent ones throughout the text).

·

Basic reporting

#Title:
Metabolomics window into the role of acute kidney injury after coronary artery bypass grafting in chronic kidney injury among diabetes mellitus patients

Authors included diabetic CKD patients? The mean eGFR was 93 ml/min/1.73^2. All of them had persistent proteinuria? Authors mentioned they included T2DM patients with normoalbuminuria.

Title should be corrected.

Experimental design

They used AKIN criteria for defining AKI, but the KDIGO guideline is a recent one. Please use this criteria.

Validity of the findings

No comment

Additional comments

Most of issues are well addressed.

---

## Round 0.3 · accepted · Accept

The authors have satisfactorily addressed all the reviewers' concerns.

·

Basic reporting

No comment

Experimental design

No comment

Validity of the findings

No comment

Additional comments

No comment